# Migraine monoclonal antibodies against CGRP change brain activity depending on ligand or receptor target – an fMRI study

**Hauke Basedau, Lisa-Marie Sturm, Jan Mehnert, Kuan-Po Peng, Marlene Schellong, Arne May***

Department of Systems Neuroscience, University Medical Center Hamburg-Eppendorf, Hamburg, Germany

## Abstract

**Background:** Monoclonal antibodies (mAbs) against calcitonin gene-related peptides (CGRP) are novel treatments for migraine prevention. Based on a previous functional imaging study which investigated the CGRP receptor mAb (erenumab), we hypothesized that (i) the CGRP ligand mAb galcanezumab would alter central trigeminal pain processing; (ii) responders to galcanezumab treatment would show specific hypothalamic modulation in contrast to non-responders; and (iii) the ligand and the receptor antibody differ in brain responses.

**Methods:** Using an established trigeminal nociceptive functional magnetic imaging paradigm, 26 migraine patients were subsequently scanned twice: before and 2–3 weeks after administration of galcanezumab.

**Results:** We found that galcanezumab decreases hypothalamic activation in all patients and that the reduction was stronger in responders than in non-responders. Contrasting erenumab and galcanezumab showed that both antibodies activate a distinct network. We also found that pre-treatment activity of the spinal trigeminal nucleus (STN) and coupling between the STN and the hypothalamus covariates with the response to galcanezumab.

**Conclusions:** These data suggest that despite relative impermeability of the blood-brain barrier for CGRP mAb, mAb treatment induces certain and highly specific brain effects which may be part of the mechanism of their efficacy in migraine treatment.

**Funding:** This work was supported by the German Ministry of Education and Research (BMBF) of ERA-Net Neuron under the project code BIOMIGA (01EW2002 to AM) and by the German Research Foundation (SFB936-178316478-A5 to AM). The funding sources did not influence study conduction in any way.

**Clinical trial number:** The basic science study was preregistered in the Open Science Framework (https://osf.io/m2rc6).

*For correspondence:
a.may@uke.de

## Editor's evaluation

Antibodies to CGRP or to the CGRP receptor are now approved for the management of migraine. Interestingly, although antibodies are presumed not to penetrate the blood-brain barrier, consistent with findings previously shown for antibodies directed against the receptor, the authors now report that galcanezumab, which targets the CGRP peptide, decreases hypothalamic activation, more so in responders. They suggest that these CNS actions may underlie part of the efficacy of these drugs in the management of migraine.

## Introduction

Since the introduction of the new generation of 'migraine-specific drugs', the monoclonal antibodies (mAbs) against the calcitonin gene-related peptide (CGRP-L-mAb) and its receptor (CGRP-R-mAb), their exact mode of action has been intensely studied. Whereas a central site of action is, due to limited permeability of the blood-brain barrier for large molecules, considered to be of minor relevance (*Johnson et al., 2019*), it is predominantly thought that CGRP mAb act peripherally by modulating the interaction between C-type and Aδ-type sensory neurons in the trigeminal ganglion and trigeminal nerve fibers (*Edvinsson et al., 2018*; *Edvinsson et al., 2019*), and that CGRP mAb may block CGRP responses in the dura mater (*Edvinsson et al., 2018*; *Melo-Carrillo et al., 2017*; *Edvinsson, 2017*).

In our recent study in migraine patients, we demonstrated that administration of the CGRP mAb erenumab results in a diminished activation of the thalamus, the insular cortex, the periaqueductal gray, and the secondary somatosensory cortex. Only responders showed less activation in the hypothalamus whereas non-responders did not, which suggests that CGRP mAb have direct or indirect central effects (*Ziegeler et al., 2020*).

The mode of action may however differ between different CGRP mAbs (*Ziegeler and May, 2020*). We focused on this question and studied 26 migraine patients before and after administration of the CGRP mAb galcanezumab using the same event-related functional magnetic resonance imaging (fMRI) and trigeminal nociceptive paradigm that we have used in the erenumab study (*Stankewitz et al., 2010*; *Schulte et al., 2016*). Given the results of the erenumab study, we hypothesized that (i) galcanezumab would alter central trigeminal pain processing in migraine patients, (ii) responders to galcanezumab treatment would show specific hypothalamic modulation in contrast to non-responders. We preregistered the study and the hypothesis (https://osf.io/m2rc6).

## Materials and methods

### Preregistration

This study was preregistered on May 12, 2020 (title 'Galcanezumab in the migraine brain – fMRI') on the Open Science Framework (https://osf.io/m2rc6).

### Patient consent

The study was approved by the local ethics committee in Hamburg, Germany (PV 5964) and was conducted in accordance with the Declaration of Helsinki. Written informed consent was obtained before initiation of the first study session. Participants could discontinue the study at any time.

### Patients

Participants were recruited by headache specialists of the headache and facial outpatient clinic of the University Medical Center Hamburg-Eppendorf, Germany.

All study participants fulfilled the diagnosis of migraine (chronic and episodic) according to ICHD-3 criteria (*No authors listed, 2018*) and kept a headache diary. Drug-naive participants to any CGRP-antibody treatment were eligible when a therapy with galcanezumab 240 mg (loading dose) was planned following national treatment guidelines (*Diener et al., 2020*). In case of additional preventative treatments, the dose of this treatment must have been unchanged for at least 3 months prior to study participation and was not allowed to change during the study. Comorbid primary or secondary headache disorders (including medication overuse headache) were excluded and none of the participants suffered from severe comorbid psychiatric, neurological, or other somatic conditions. Two patients reported a mild restless legs syndrome.

### Experimental paradigm

Patients were asked to attend two fMRI scanning sessions before and after the first administration of galcanezumab. Both visits took place approximately 3 weeks apart (pharmacokinetic drug peak blood level) and followed the same protocol. Following the first scan, the loading dose of galcanezumab 240 mg was administered subcutaneously by the patient under the prior instruction of a headache specialist.

During the experiment, which has been described in detail previously (*Stankewitz et al., 2010*; *Schulte et al., 2016*), participants received 15 repetitive painful trigeminal stimulations by administering gaseous ammonia (mixed in a stream of air) into the left nostril and 15 puffs of air, 15 trials of rose scent, and 15 repetitive visual stimuli as control conditions. All stimuli were presented in a pseudo-randomized order, with painful trigeminal stimuli not presented in immediate succession. After each trial, participants were asked to rate the intensity as well as the unpleasantness on a numeric rating scale from 0 to 100, each of which was tested with a paired t-test if the criteria of the Gaussian distribution were fulfilled or otherwise with a Wilcoxon signed-rank test. Standardized headache calendars were collected at visit 1, visit 2, and in a follow-up interview by telephone and email after a total of 3 months of therapy (Figure 2A).

## MRI acquisition

All images were recorded at a Siemens PRISMA 3T MR system (Siemens, Erlangen, Germany) using a 64-channel head coil. During the experimental protocol, 1095 functional images were acquired for each subject and for each session using an echoplanar imaging sequence optimized for blood-oxygenation level-dependent (BOLD) brainstem imaging (*Schulte et al., 2016*): voxel size $1.3 \times 1.3 \times 2.5$ mm$^3$, 38 axial slices (no gap), repetition time 2.64 s, echo time 28 ms, flip angle 80°, GRAPPA acceleration mode, field of view readout 216 mm, phase partial Fourier 7/8, two saturation pulses were added anterior and posterior to the target volume, which covered the whole volume from the corpus callosum to the foramen magnum (MNI z-range 25–72). Simultaneously, we recorded pulse and breathing (Expression, Philipps, Best, The Netherlands) to correct for cardiovascular artifacts.

Functional imaging was followed by field mapping MRI sequence (repetition time 0,8 s, echo time 1: 5.51 ms, echo time 2: 7.97 ms, flip angle 40°, field of view readout 215 mm) and a high-resolution magnetization-prepared rapid gradient echo sequence image (voxel size 1 mm$^3$, repetition time 2.3 s, echo time 2.98 ms, flip angle 9°, field of view 256 mm$^2$, 240 axial slices gap 50%).

## Preprocessing

In general, the preprocessing followed the established basis for functional imaging preprocessing steps consisting of: denoising functional images (spatially adaptive non-local mean filter) (*Manjón et al., 2010*), realignment and unwarping by the aforementioned field maps, slice time correction, co-registration to the anatomical images, and transformation into Montreal Neurological Institute (MNI) space as implemented in SPM12 (Wellcome Trust Center for Neuroimaging, London, UK). Functional images were then smoothed using a 4 mm$^3$ full width at half maximum Gaussian kernel.

## Statistical analysis

A general linear model (GLM) analysis was used within each participant providing β-estimates which were used for group statistical analysis. These β-values were calculated for each voxel and signify the condition-specific neuronal activity. Therefore, we were using a hemodynamic response function to model all four stimulus conditions (ammonia, rose, air puff, visual) and three confound conditions (key press/assessment, attention task, anticipation phase) by convolving their onsets and durations and applying them as regressors in the GLM. For further correction of movements that were not intercepted by the realignment processing, we included the six movement regressors provided in the realign and unwarp step mentioned earlier. For physiological noise correction, we included an additional 18–20 regressors extracted from each participants' breath and pulse signals using the approach described by *Deckers et al., 2006*. For the main effect, the results were corrected for multiple comparisons (family-wise error corrected, $p<0.05$), for the sub-analysis with a strong a priori hypothesis (see preregistration and *Ziegeler et al., 2020*), we calculated paired and independent one-sided t-test as implemented in the SPM toolbox and used an uncorrected statistical threshold of $p<0.001$.

## Arterial spin labeling

As galcanezumab is potentially a vasoactive drug, arterial spin labeling (ASL) was also recorded to exclude that any BOLD changes were due to general changes in cerebral blood flow (CBF). To cover the blood flow of the entire brain, we performed separate measurements of the brainstem and the cerebrum. The ASL sequence used pulsed ASL recorded with 91 repetitions in 17 slices with a TR of 2.6 s (TE 12 ms, 90° flip angle, bolus duration 1800 ms, inversion time 700 ms, PICORE Q2T perfusion

mode, voxel size 2 × 2 × 5 mm³). The relative CBF maps calculated by the scanner software were co-registered to the anatomical image, warped into MNI space using the transformation calculated on the anatomical image, and smoothed using a 12 mm isotropic Gaussian kernel again using SPM12 (Wellcome Trust Center for Neuroimaging, London, UK). Results were corrected for multiple comparisons (family-wise error corrected, $p<0.05$, $T>6.1$, df = 25).

### Effect of trigeminal stimulation

First, both visits were pooled to observe the effect of the trigeminal nociceptive ammonia stimulation (main effect). A statistical threshold of $p<0.05$ (FWE corrected for multiple comparisons) was used for a one-sided, independent t-test.

### Effect of galcanezumab on trigeminal stimulation

To estimate significant differences before vs. after administration of galcanezumab, a one-sided, paired t-tests in each voxel was performed using *SPM12, 2014* for the contrast of pain > control at visit 1 vs. pain > control at visit 2 (further classified as [ammonia-air puffs]$_{visit 1}$ > [ammonia-air puffs]$_{visit 2}$ and [ammonia-air puffs]$_{visit 1}$ < [ammonia-air puffs]$_{visit 2}$). A statistical threshold of $p<0.001$ (uncorrected) was used, since we had a strong a priori hypothesis, see preregistration and *Ziegeler et al., 2020*.

### Differences between galcanezumab (CGRP-L-mAb) and erenumab (CGRP-R-mAb)

As this study followed a previously conducted study on the CGRP receptor mAb erenumab (*Ziegeler et al., 2020*), we also compared the two effects caused by these medications. To ensure direct comparability, we also included a subgroup consisting of patients with the same migraine phase in the analyses and compared the respective contrasts ([ammonia-air puffs]$_{visit 1}$ vs. [ammonia-air puffs]$_{visit 2}$) in a one-sided, two-sample t-test. For this analysis, the existing erenumab data sets were reassessed and included in above-mentioned analysis routine. A statistical threshold of $p<0.001$ (uncorrected) was used, since we had a strong a priori hypothesis, see preregistration and *Ziegeler et al., 2020*.

### Therapy prediction analysis

Following the protocol of the previous study (*Ziegeler et al., 2020*), we defined and preregistered 'being a responder' by showing a 30% reduction in headache frequency after 3 treatment months. Since a maximal response was not to be expected at 3 weeks, we used the accepted time span of 3 months to calculate responders (*Goadsby et al., 2019*). However, we asked all patients at the time of the second scan whether they felt the medication to be effective and the number of patients who answered this subjective question positively, corresponded with the 30% response at 3 months (two-tailed Pearson's correlation, $r=0.778$, $p<0.001$). To detect possible predictors of treatment outcome in the processing of trigemino-nociceptive stimuli, we used the visit 1 contrast [ammonia-air puffs]$_{visit 1}$ and co-varied the model with the individual response after 3 months (in % reduction of monthly headache days [MHD]) in all patients (n=26) in an one-sided, independent t-test as implemented into the SPM toolbox. An additional nuisance covariate of migraine phase was used to eliminate possible variance caused by the presence of headache. A statistical threshold of $p<0.001$ (uncorrected) was used for hypothesis generation.

### Alterations in functional connectivity

Galcanezumab-induced functional connectivity changes were estimated for the whole group (n=26) by psychophysiological interaction (PPI) analysis (*Friston et al., 1997*), using the region of interest (i.e. spinal trigeminal nucleus [STN]) resulting from the previous analysis as a starting point and contrasting ammonia (nociceptive input) and air puffs (control condition) between the two visits as before. A statistical threshold of $p<0.001$ (uncorrected) was used in the one-sided, independent t-test, since we had a strong a priori hypothesis, see preregistration and *Schulte et al., 2016*; *Schulte and May, 2016*.

**Table 1.** Patient characteristics.

| Patient characteristics | Galcanezumab | *Erenumab* |
|---|---|---|
| Number | 26 | 27 |
| Female, % (n) | 96 (25) | 81 (22) |
| Age, mean ± SD (range), in years | 37.81±12.11 (21–60) | 39.1±12.2 (22–60) |
| Disease duration, mean ± SD (range), in years | 19.77±12.43 (2–50) | 21.7±11.2 (6–43) |
| Headache frequency, mean ± SD (range), days/month<br>EM: mean ± SD (range), days/month<br>CM: mean ± SD (range), days/month | 16.71±8.92 (4–30)<br>10.28±2.94 (4–14)<br>25.48±6.32 (16–30) | 20.3±8.3 (8–30)<br>12.08±2.07 (8-14)<br>26.8±4.71(20–30) |
| Migraine with and without aura, n | 7 | 9 |
| Migraine without aura, n | 19 | 18 |
| Chronic migraine (ICHD-3), % (n) | 42 (11) | 56 (15) |
| Episodic migraine (ICHD-3), % (n) | 58 (15) | 44 (12) |
| Same headache state – episodic, % (n) | 40 (6) | 29 (5) |
| – chronic, % (n) | 60 (9) | 71 (12) |

Abbreviation: ICHD-3=International Classification of Headache Disorders, 3rd edition, EM: episodic migraine, CM: chronic migraine.

## Results

### Patient characteristics

Between May 2020 and January 2021, a total of 29 patients were enrolled in the study. One patient administered 120 rather than 240 mg loading dose and technical issues in two cases led to exclusion. Twenty-six patients (19 without aura and 7 with aura) (***No authors listed, 2018***) were included. Of these, 11 patients fulfilled the formal criteria for chronic migraine and 15 patients for episodic migraine, based on headache diaries. A total of 13 patients were taking medications other than hormonal contraception or dietary supplements for other indications: Eight patients took (because

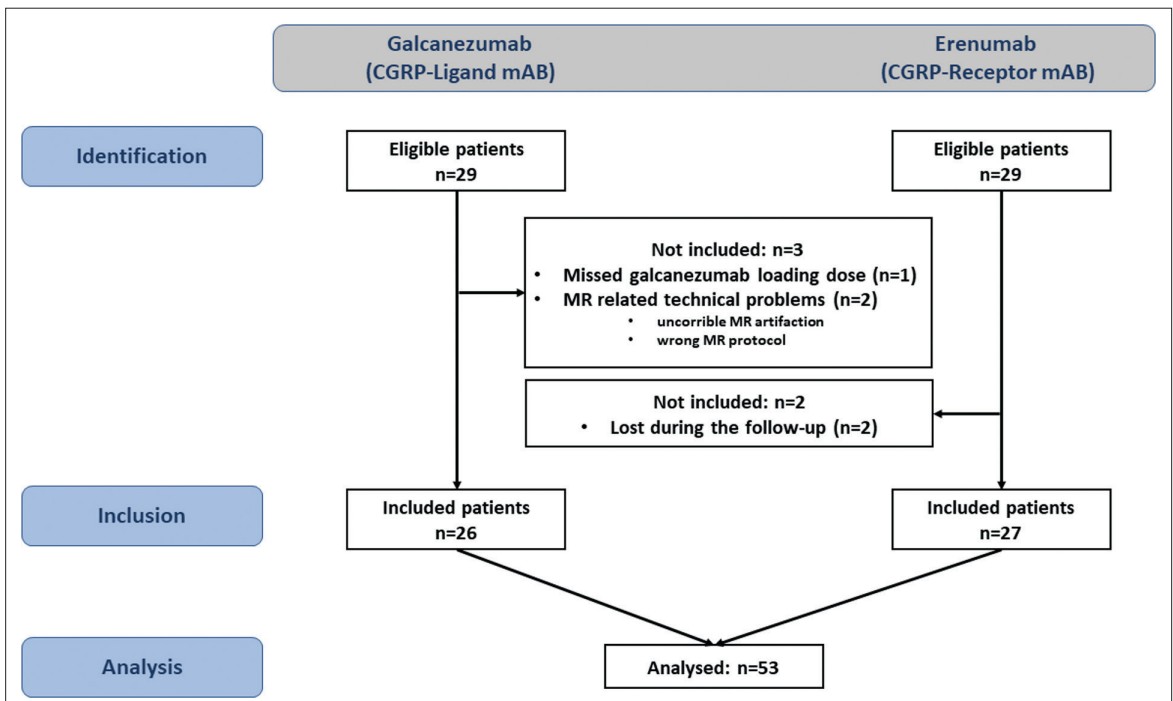

**Figure 1.** Study flowchart. On the left side the galcanezumab cohort, on the right side the erenumab cohort.

of migraine) antidepressant medication (selective serotonin reuptake inhibitor, serotonin and norepinephrine reuptake inhibitor, or tricyclic antidepressant), two patients took antihypertensive medication (metoprolol, bisoprolol), and another two anticonvulsive drugs (topiramate); two patients were taking L-dopa due to restless legs syndrome. Each of the concomitant medications was used as a covariate in the imaging analysis but had no effect on the results. Patient characteristics are shown in *Table 1* and the study flowchart in *Figure 1*.

## Headache state group/covariation

Because headache state can have a major impact on the results (*Schulte et al., 2017*), we followed the same protocol as for the erenumab study (*Ziegeler et al., 2020*) and defined a subgroup with the same headache condition during both sessions (n=15), that is, headache-free or headache on both visit days to control for possible changes due to headache phase. Additionally, the headache state was also modulated as a covariate of nuisance (noted below in each respective analysis).

Fourteen patients had headache on visit 1, whereas only 11 patients had headache on visit 2, resulting in a total of 15 patients (episodic n=6, chronic n=9) who had equal headache (n=7) or no headache (n=8) on both dates (for erenumab study: 17 patients [episodic n=5, chronic n=12] with headache [n=11] and no headache [n=6] on both dates). In patients with headache, the intensity of headache did not differ between both visits.

## Responder group

Ten out of 26 (38%) of the included patients were responders (50% decrease in MHD), while 18 out of 26 (69%) showed a decrease of 30% in MHD and were defined as responders since this group consisted (following the European prescribing instructions) of difficult-to-treat migraine patients and also according to previous studies (*Ziegeler et al., 2020*; *Overeem et al., 2022*) and the preregistration.

**Table 2.** Details of the statistical results of the functional magnetic resonance imaging (fMRI) analyses comparing before and after treatment as well as responders and non-responders.

| Anatomical region | Cluster size (voxels), n | T value | MNI coordinates (x,y,z) |
|---|---|---|---|
| *(A) Visit 1>visit 2, all participants (n=26, contrast [ammonia-air puffs]*visit1 *> [ammonia-air puffs]*visit2*,threshold: p<0.001 [uncorrected], T>3.45, minimum cluster extent 25 voxels, df = 25)* | | | |
| R hypothalamus | 49 | 4.99 | 8, –14, –7 |
| R cerebellum | 48 | 5.0 | 20, –40, –25 |
| R cerebell. vermis | 52 | 4.09 | 3, –67, –34 |
| *(B) Responder (n=8)>non-responder (n=7) with same headache state on both days (contrast [ammonia-air puffs]*visit1 *> [ammonia-air puffs]*visit2*, threshold p<0.001 [uncorrected], T>3.85, minimum cluster extent 10 voxels, df = 14)* | | | |
| R inf. parietal lobule | 91 | 9.17 | 58, –44, 21 |
| L precentral gyrus | 26 | 5.99 | –48, –11, 11 |
| L parahippocampal gyrus | 12 | 5.06 | –18, –11, –19 |
| L superior temporal gyrus | 13 | 4.76 | –36, 15,–26 |
| L inf. parietal lobule | 17 | 4.7 | 46, –42, 22 |
| R lentiform nucleus | 16 | 4.67 | 24, –2, 4 |
| R parahippocampal gyrus | 27 | 4.63 | 27, –28, –12 |
| L insula | 26 | 4.56 | –45, –27, 19 |
| L hypothalamus | 13 | 4.47 | –6, –17, –6 |

(A) Main findings of trigeminal pain processing alterations after administration of the CGRP-ligand monoclonal antibody galcanezumab. The contrast shows more neuronal activity on visit 1 than on visit 2, therefore decrease driven by galcanezumab and was tested with a one-way paired t-test. (B) Subgroup analysis of patients having the same migraine/headache state on both visits (ictal and ictal or interictal and interictal). Main findings specific for being responder (responder showing a higher decrease after galcanezumab than non-responder) tested with a one-way independent sample t-test. Note that left-right activation patterns in near proximity twin structures such as the hypothalamus are not necessarily side-locked (a left activation excludes a right activation) but may be due to statistical thresholding.

## Behavior

The behavioral data between visit 1 and visit 2 for intensity and unpleasantness ratings of the nociceptive stimulation was not significantly different ($p_{*Bonferroni}$ = 0.094 with T = 2.4, df = 25, average difference 6.62/100, 95% confidence interval [CI] –0.90 to 11.58 for intensity; $p_{*Bonferroni}$ = 0.156 with T = 2.18, df = 25, average difference 5.2/100, 95%, CI –0.28 to 10.12 for unpleasantness ratings) tested by a two-sided, paired t-test (corrected for multiple comparisons by Bonferroni correction) with given normal distribution. The same was true for the other conditions: control (air puff), olfactory input (rose scent), and visual stimulation.

## Arterial spin labeling

ASL was performed in a two-step whole-brain protocol and showed no changes in relative CBF (corrected for multiple comparisons [family-wise error corrected, $p<0.05$, T>6.1, df = 25]).

## Effect of trigeminal stimulation

The (pooled) main effect of trigeminal nociceptive stimulation revealed cortical and subcortical structures mediating central pain/salience processing, like ipsilateral STN, contralateral thalamus, insula, and cerebellum with a statistical threshold of $p<0.05$, corrected for multiple comparisons (FWE).

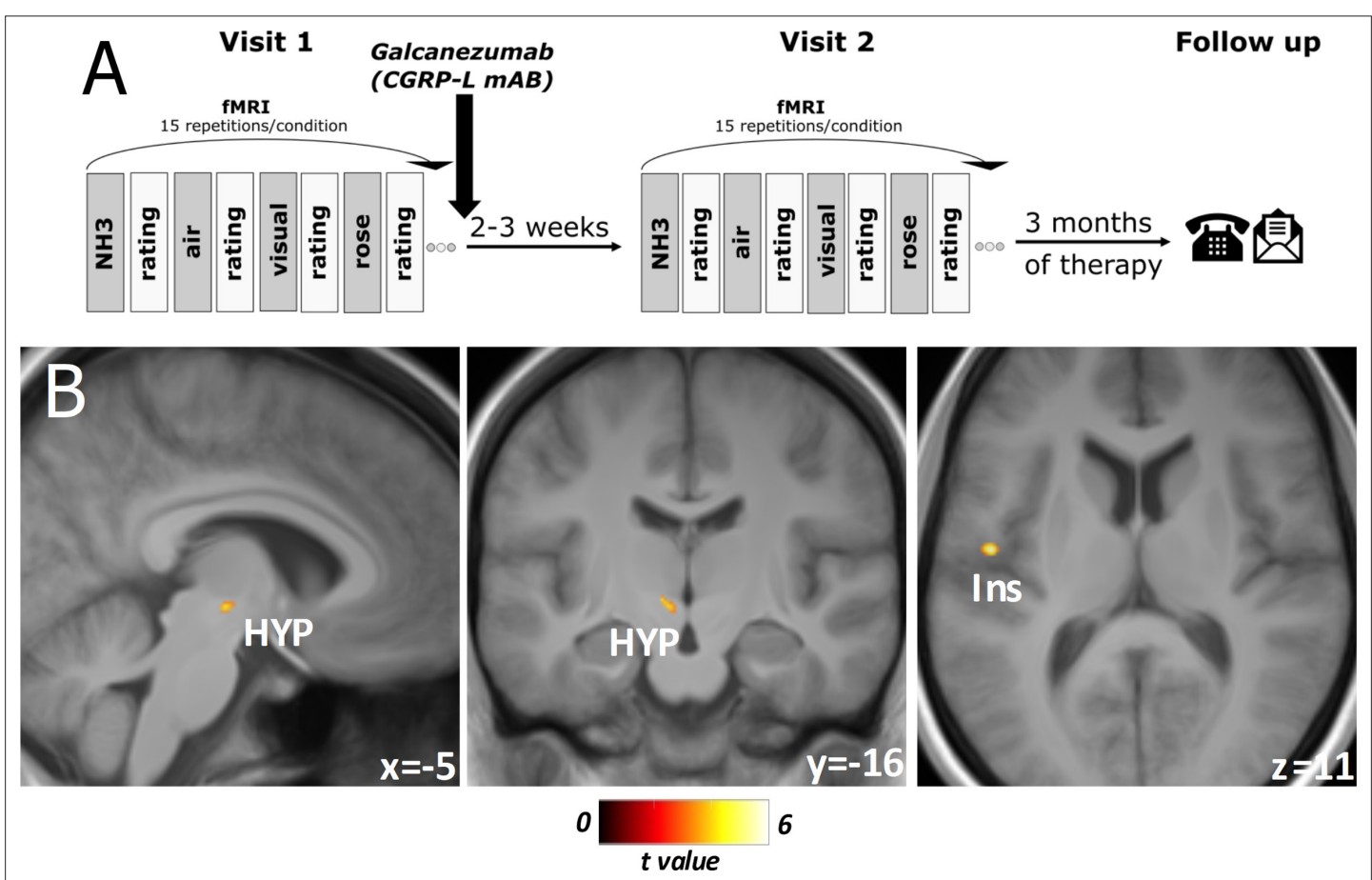

**Figure 2.** Experimental design and responder-specific main findings. (**A**) Flowchart of the experimental design. At both visits, objective rating of pain intensity and unpleasantness on a numeric rating scale (NRS), session-specific questionnaire: current headache (yes/no), strength of headache (NRS), headache frequency (per week), date of last headache. Three months of total therapy duration, patients were contacted via phone and/or email to acquire their headache diaries. (**B**) Responder-specific decreased neuronal activity during nociception at visit 2 compared to visit 1 in subgroup analysis of migraine-phase matched patients (n=15). Data is shown at statistical threshold of p<0.001 and a minimum cluster size of 10 voxels, masked with gray and white matter study template where a higher t-value means a stronger decrease of activity after treatment for responders in comparison to non-responders.

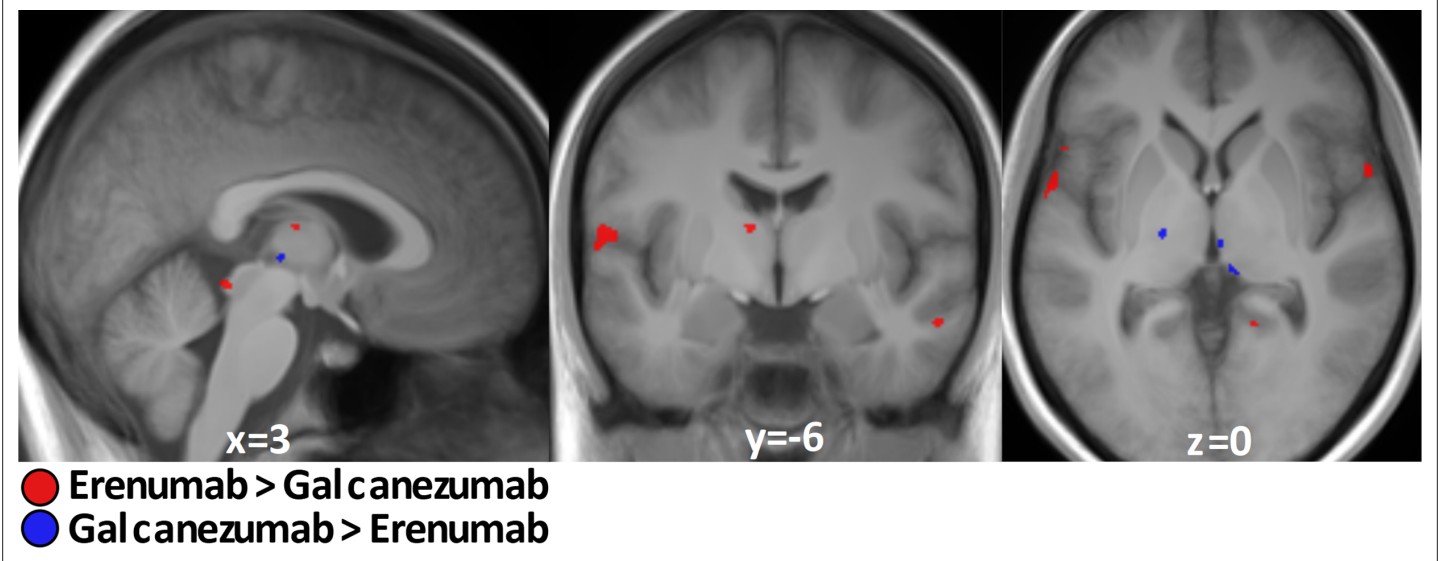

**Figure 3.** Galcanezumab vs erenumab. Erenumab (red)/galcanezumab (blue)-specific decreased neuronal activity during nociception at visit 2 compared to visit 1 in subgroup analysis of migraine-phase matched patients (n=15 vs. 17). Data is shown at statistical threshold of p<0.001 and a minimum cluster size of 10 voxels.

### Effect of galcanezumab on trigeminal stimulation

Galcanezumab treatment (n=26; [ammonia-air puffs]$_{visit1}$>[ammonia-air puffs]$_{visit2}$) was associated with a decrease in activation in the right hypothalamus, right cerebellum, and cerebellar vermis (*Table 2A*), with a statistical threshold of p<0.001 (uncorrected) and a minimum cluster extension of 25 voxels, following previous studies with previously generated and preregistered hypotheses. The inverse contrast ([ammonia-air puffs] $_{visit1}$<[ammonia-air puffs]$_{visit2}$) showed no significant effects.

Comparing responders vs. non-responders in a subgroup consisting only of patients within the same migraine phases (visit 1 and visit 2 either with or without headache, n=15; eight responders), we found a responder-specific decrease in BOLD activity in central areas such as the inferior parietal lobule/operculum, the left insula, the parahippocampal gyrus, as well as the hypothalamus (*Figure 2B*, *Table 2B*). Non-responders only showed a specific decrease in the cerebellar tonsils (p<0.001, T=4.81, $k_E$ = 66 voxels, −38, –62, –49 [x, y, z/mm]). Using the 50% responders showed the same results but at lower exploratory threshold levels since only 10 out of 26 of the included patients were 50% responders.

### Galcanezumab (CGRP-L-mAb) vs. erenumab (CGRP-R-mAb)

Exclusion and inclusion criteria between this study and the previous erenumab study (*Ziegeler et al., 2020*) were the same for reasons of comparability. Patient characteristics (sex, age, migraine type) and behavioral ratings (pain severity and unpleasantness of stimuli) were also comparable and not significantly different. It is therefore feasible to directly compare both studies in the same analysis and we focused on the migraine-phase equal subgroup analysis (n=15 galcanezumab vs. n=17 erenumab) to eliminate possible influences of different migraine phases. Erenumab specifically decreased activation in the operculum, insula, thalamus, and cerebellum (*Figure 3*, *Table 3A*), whereas galcanezumab led to a specific decrease in activity in the hypothalamic area, left thalamus, as well as in pontine region (*Figure 3*, *Table 3B*). The inverse contrast showed no results.

### Therapy prediction analysis

Contrasting the activation maps on visit 1 [n=26; ammonia-air puffs]$_{visit1}$ with the response (% reduction MHD) 3 months later showed that the STN highly co-varied with the response (*Figure 4A*, *Table 4A*) at a statistical threshold of p<0.001. Given that the STN has a central role in the ascending pathway for processing trigemino-nociceptive stimuli, we extracted the parameter estimates from the F-contrast in the peak voxel of the STN and correlated it with the response to galcanezumab (% reduction

**Table 3.** Details of the statistical results of the functional magnetic resonance imaging (fMRI) analyses comparing calcitonin gene-related peptides (CGRP) receptor monoclonal antibody erenumab and CGRP ligand monoclonal antibody galcanezumab.

| Anatomical region | Cluster size (voxels), n | T value | MNI coordinates (x,y,z) |
|---|---|---|---|
| *(A) Erenumab >Galcanezumab (Visit 1>visit 2, migraine-phase equal subgroup [n=15 vs. n=17], contrast [ammonia-air puffs]_{visit1} > [ammonia-air puffs]_{visit2},threshold: p<0.001 [uncorrected], T>3.39, minimum cluster extent 10 voxels, df = 30)* | | | |
| L operculum | 416 | 5.61 | −60, −5, 8 |
| L cerebellum | 122 | 5.59 | −16, −65, −22 |
| R cerebellum | 207 | 5.23 | 42, −51, −32 |
| R supramarginal gyrus | 37 | 5.11 | 64, −45, 23 |
| R thalamus | 67 | 5.0 | 6, −15, 11 |
| R hippocampus | 74 | 4.88 | 21, −32, −10 |
| L thalamus | 53 | 4.71 | −9, −7, 11 |
| L temporal pole/insula | 202 | 4.5 | −54, 17, −1 |
| L superior temporal gyrus | 55 | 4.42 | −55, −33, 16 |
| Locus coeruleus | 30 | 4.41 | 4, −37, −9 |
| R insula | 82 | 4.37 | 47, 14, −7 |
| R operculum | 19 | 4.15 | 62, −2, 7 |
| R middle temporal gyrus | 21 | 4.07 | 55, −6, −22 |
| R lingual gyrus | 11 | 3.9 | 16, −49, 0 |
| *(B) Galcanezumab >Erenumab (Visit 1>visit 2, migraine-phase equal subgroup [n=15 vs. n=17], contrast [ammonia-air puffs]_{visit1} > [ammonia-air puffs]_{visit2},threshold: p<0.001 [uncorrected], T>3.39, minimum cluster extent 10 voxels, df = 30)* | | | |
| Pons | 40 | 4.09 | −1, −18, −38 |
| R substantia nigra | 13 | 4.01 | 9, −14, −10 |
| L thalamus | 13 | 3.93 | −18, −15, 0 |
| R hypothalamus | 12 | 3.89 | 4, −19, 0 |

Main findings of trigeminal pain processing alterations driven by the administration of the CGRP receptor monoclonal antibody erenumab in contrast to the CGRP ligand monoclonal antibody galcanezumab in the migraine-phase equal subgroup. (A) Erenumab-specific decrease in neuronal activity/BOLD across the two visits. (B) Galcanezumab-specific decrease in neuronal activity/BOLD across the two visits. Both results stem from one-sided independent t-tests.

of MHD). The Pearson's correlation showed a significant positive relationship between visit 1 STN activation and the response of each participant (two-tailed Pearson's correlation, r=0.63, p<0.001, *Figure 4B*).

## Alterations in functional connectivity

In the PPI connectivity analysis (n=26), we used the STN cluster found in the therapy prediction analyses and found a decrease in connectivity of the STN on visit 2 with the hypothalamus and also in the superior temporal gyrus (*Figure 4C*, *Table 4B*). Connectivity increases with the STN were found to areas of the cerebellum, middle temporal gyrus, and insula.

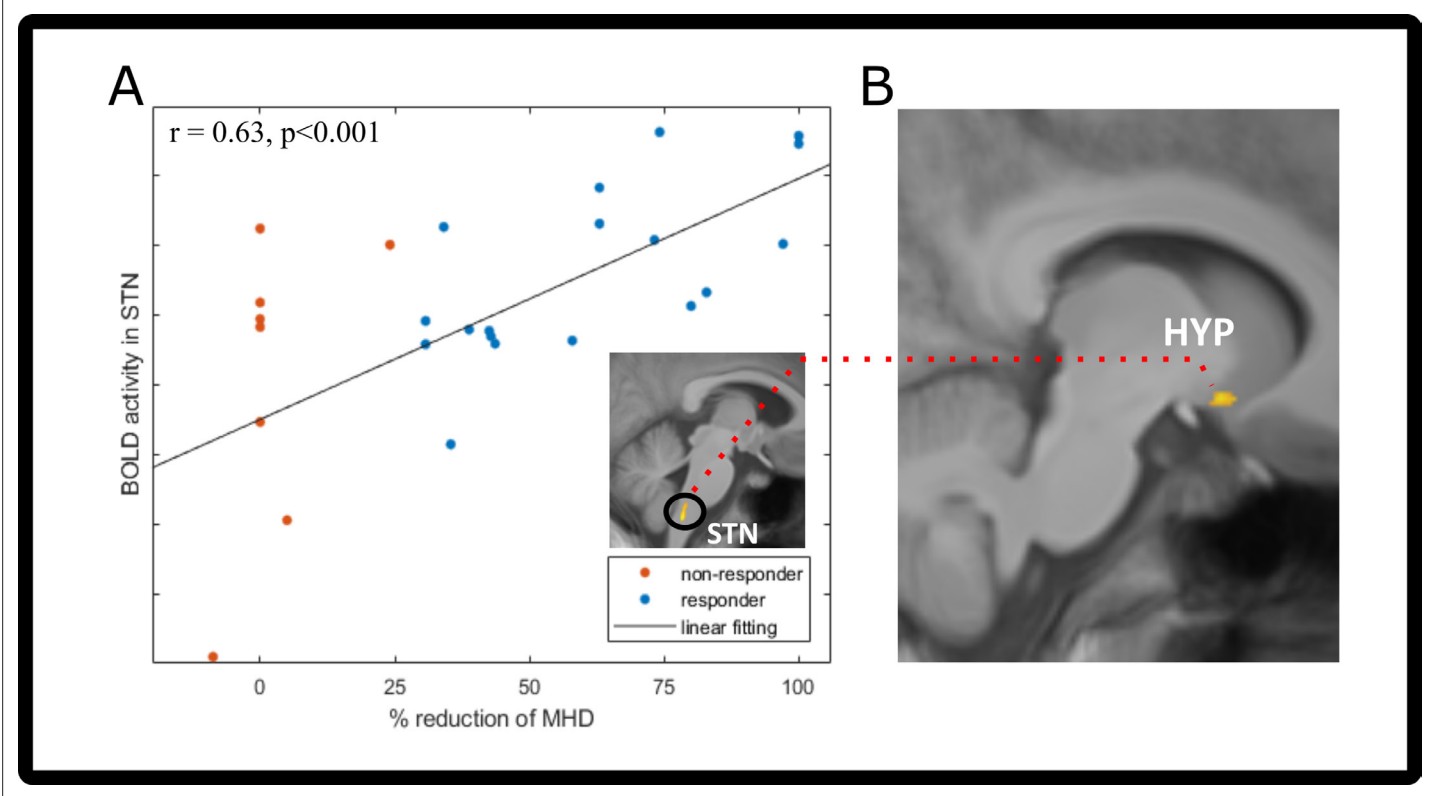

**Figure 4.** Prediction/functional connectivity changes after galcanezumab administration. (**A**) Two-tailed Pearson's correlation of visit 1 spinal trigeminal nucleus (STN) neuronal activity (before administration of galcanezumab) and response (in % change of MHD). Responder (>30% reduction) are marked in blue; non-responders (<30% reduction) are colored red. STN activation is shown in the image above the legend – visit 1 trigeminal-nociceptive activation (ammonia-air puffs) covaries with response (in percent reduction of monthly headache days [MHD]). Data is shown at statistical threshold of p<0.001 and a minimum cluster size of 10 voxels. (**B**) Results of the psychophysiological interaction (PPI) analysis, showing the decreased connectivity between STN (seed region [A]) and hypothalamus. Data is shown at statistical threshold of p<0.001.

## Discussion

Our functional imaging data suggest that galcanezumab therapy is associated with a decrease of hypothalamic activation in all patients following galcanezumab therapy, which is even more decreased in responders. This is in line with our previous study in erenumab patients (*Ziegeler et al., 2020*),

**Table 4.** Statistical results of the functional magnetic resonance imaging (fMRI) analyses regarding the coherence with the reduction of headache days after treatment and changes in functional connectivity.

| Anatomical region | Cluster size (voxels), n | T value | MNI coordinates (x,y,z) |
|---|---|---|---|
| *(A) Covariation of monthly headache day reduction (in % reduction) in the contrast (ammonia-air puffs) $_{visit1}$, all participants (n=26) threshold: p<0.001 [uncorrected], T>3.48, minimum cluster extent 25 voxels, df = 25* | | | |
| R middle temporal gyrus | 110 | 4.9 | 49, –8, –14 |
| STN | 83 | 4.65 | 4, –42, –53 |
| *(B) PPI analysis of STN (region of interest 4, –42,–53), all participants (n=26) in the contrast (ammonia-air puffs) $_{visit1}$> [ammonia-air puffs]$_{visit2}$, threshold: p<0.001 [uncorrected], T>3.45, minimum cluster extent 25 voxels, df = 25* | | | |
| L superior temporal gyrus | 62 | 5.57 | –52, –47, 18 |
| R hyopthalamus | 52 | 4.24 | 10, 5, –12 |

(A) Main findings of the covariation of the response to galcanezumab by the activation pattern of neuronal activity at visit 1 (before administration of galcanezumab). This result was gained by introducing the covariate of reduction in headache days into a one-sided, one-sample t-test. (B) Results from the psychophysiological interaction analysis of the above-mentioned STN activation.

where responders also showed decreased hypothalamic activity in comparison to non-responders. Unlike non-responders to galcanezumab in this study, we did not find decreased hypothalamic activation in non-responders to erenumab. Since this could be due to threshold effects (i.e. erenumab also reduced hypothalamic activity but at a lower statistical threshold), we compared both studies directly in the same analysis. This comparison showed that the hypothalamic decrease following galcanezumab is indeed specific to galcanezumab as it was not present in erenumab patients. Of note, the other comparison in this medication-specific analysis showed that the general effects seen after erenumab medication (*Ziegeler et al., 2020*), that is, a decrease in activation in the thalamus, right middle temporal gyrus, right lingual gyrus, and left operculum are specifically only found in the erenumab-treated patients and not after galcanezumab therapy. It is important to state that these findings are related to the response to a trigeminal-nociceptive stimulus, and not some general difference in brain state such as resting state changes for example.

Our findings suggest that the antibodies to the ligand and the receptor exhibit a similar modulatory change in the hypothalamic area if patients respond to treatment, whereas the general change due to administration of the medication is different and specific for each antibody type. Since the perception of nociceptive input elicited by our paradigm was not significantly different between the two sessions, neither in responders nor in non-responders or within the whole cohort, an external influence on the central processing of these stimuli can be excluded.

The hypothalamus plays a central role in migraine pathophysiology (*Alstadhaug, 2009Stankewitz et al., 2010*; *Overeem et al., 2022*). Studies in the rhesus monkey confirmed the expression of CGRP receptors in these areas (*Eftekhari et al., 2016*). However, a direct effect of monoclonal CGRP antibodies in the central nervous system in these areas is unlikely, because the blood-brain barrier is relatively impermeable to substances of this large molecular mass (*Johnson et al., 2019*; *Noseda et al., 2020*). However, the hypothalamus, and indeed the entire central nervous system, is subject to cyclic changes in CGRP concentrations in migraine patients which has been shown in cubital (*Greco et al., 2020*) and also plasma cranial venous blood analyses (*Goadsby et al., 1990*). Since small amounts of CGRP can cross the blood-brain barrier (*Johnson et al., 2019*), one could argue that central nervous (hypothalamic) receptors are subsequently stimulated by increased cranial CGRP levels during a migraine attack. Systemic administration of galcanezumab binds free circulating CGRP (*Kielbasa and Helton, 2019*) and this modulates the ability to cross the blood-brain barrier from the periphery to the cerebrospinal fluid compartment. Assuming intra- and extrathecal CGRP concentrations pursue an equilibrium or equal concentration gradients, such a decrease in functional CGRP due to antibodies could subsequently lead to reduced CGRP-mediated activation of hypothalamic and other brain areas that express CGRP receptors. This would account for the reduction of hypothalamic activity in all subjects as well as in the responder vs. non-responder comparison. Of note, studies showing an elevated CGRP level in migraine attacks (*Goadsby et al., 1990*) have to be seen in the light of studies showing that administration of CGRP may trigger migraine-like attacks (*Hansen et al., 2010*) but is in itself not painful (*Asghar et al., 2016*), suggesting that the elevation of CGRP levels in migraine attacks may be a consequence and not the primary trigger of the attack. That being said, erenumab elicits other modes of action (e.g. intracellular) than just blocking or inactivating the receptor (*Bhakta et al., 2021*). This could explain why we see different central responses to erenumab and galcanezumab, and it could also explain why some non-responders to CGRP-Ab (erenumab or galcanezumab) may indeed have a better response by switching to the other antibody class (*Ziegeler and May, 2020*).

Given that certain areas of the central nervous system like the area postrema or the hypothalamus are not completely protected by the blood-brain barrier (*Eftekhari et al., 2016*), one could also argue that the antibodies could still reach the hypothalamus even with intact blood-brain barrier in a very small but effective amount. This could be different for the antibody type and thus could explain why we see different central responses to erenumab and galcanezumab, and it could also explain why some non-responders to CGRP-Ab (erenumab or galcanezumab) may indeed have a better response by switching to the other antibody class (*Ziegeler and May, 2020*).

We also found that the STN, which plays a central role in the ascending trigemino-nociceptive pathway, co-varies in its activity already at visit 1 with the response, that is, the efficacy to galcanezumab. In other words, the higher the STN activity to nociceptive stimuli before administration of the drug, the more likely that galcanezumab is effective. One could argue that a sensitization of the

STN might be the missing link why some patients benefit from anti-CGRP therapy, others do not. The STN exhibits a particularly high density of CGRP due to termination of afferents from the trigeminal ganglion (*Eftekhari et al., 2016*; *Eftekhari and Edvinsson, 2011*). Recent findings on molecular brainstem and trigeminal ganglion mechanisms suggest that presynaptic CGRP-evoked intracellular mechanisms may lead to a possibly long-term modulation of signaling cascades, which in their consequence could lead to a facilitation and strengthening of (trigeminal) nociceptive transmission (*Messlinger, 2018*). However, we did not investigate sensitization levels or allodynia in our patients and explanations for this correlation between STN activity and treatment response stays speculative at the moment. It is interesting, however, that we found that galcanezumab also decreased the well-known functional connection between the STN and the hypothalamus (*Schulte et al., 2016*; *Schulte and May, 2016*; *Schulte et al., 2017*). Our data suggest that, in addition to peripheral mechanisms, the combination of decreased hypothalamic activity and a simultaneous decrease of the hypothalamic-brainstem connectivity is needed for CGRP antibodies to be effective. The question of whether this is the cause or consequence of blocking the CGRP ligand cannot be answered with our study.

Some limitations of this study have to be discussed: The number of patients (n=26) in this longitudinal design is difficult to achieve but allows only small subgroups (such as responder/non-responder). We note that the groups were big enough to find meaningful differences between groups and moreover, these findings confirmed our preregistered a priori hypothesis. One could ask the question why we defined treatment response of 30%, rather than 50%. We followed the IHS (International Headache Society) efficacy parameter 'attack frequency' although most of our patients reported a marked effect of attack severity rather than frequency reduction. In clinical practice, this means that many patients who show significant improvement in quality of life and reduction in acute medication consumption, and are therefore responders, may still have nominally a high migraine attack frequency. Therefore, because of the methodology used, the observations of this study in terms of responders may be limited to patients who are frequency responders. One could argue that only in frequency responders the hypothalamus is effectively targeted using mAbs and not in severity responders. Unfortunately, given the small group sizes of intensity vs. frequency responders, a subgroup analysis in this regard was not possible, which would have been necessary to answer this question directly.

We also note that, following national guidelines (*Diener et al., 2020*), patients receiving mAbs must have demonstrated an unsuccessful treatment effect of a least four different other preventatives and are so-called severe treatment non-responders. Both facts prompted us to focus on a 30% frequency reduction from baseline, which is sufficient to demonstrate robust differences between groups.

## Acknowledgements

This work was supported by the German Ministry of Education and Research (BMBF) of ERA-Net Neuron under the project code BIOMIGA (01EW2002 to AM) and by the German Research Foundation (SFB936-178316478-A5 to AM). The funding sources did not influence study conduction in any way.

## Additional information

### Competing interests

Hauke Basedau: HB received consulting fees from Novartis and Teva. The author has no other competing interests to declare. The other authors declare that no competing interests exist.

### Funding

| Funder | Grant reference number | Author |
| --- | --- | --- |
| Bundesministerium für Bildung und Forschung | BIOMIGA (01EW2002 to AM) | Arne May |
| Deutsche Forschungsgemeinschaft | SFB936- 178316478 - A5 | Arne May |

| Funder | Grant reference number | Author |
|--------|------------------------|--------|

The funders had no role in study design, data collection and interpretation, or the decision to submit the work for publication.

## Author contributions

Hauke Basedau, Formal analysis, Investigation, Writing – original draft; Lisa-Marie Sturm, Kuan-Po Peng, Marlene Schellong, Investigation, Writing - review and editing; Jan Mehnert, Formal analysis, Writing - review and editing; Arne May, Conceptualization, Funding acquisition, Supervision, Validation, Writing – original draft, Writing - review and editing

## Author ORCIDs

Hauke Basedau http://orcid.org/0000-0002-4605-0172
Jan Mehnert http://orcid.org/0000-0001-7088-740X
Marlene Schellong http://orcid.org/0000-0002-6499-2024
Arne May http://orcid.org/0000-0002-3499-1506

## Ethics

Human subjects: The study was approved by the local ethics committee in Hamburg, Germany (PV 5964) and was conducted in accordance with the Declaration of Helsinki. Written informed consent was obtained before initiation of the first study session.

## Decision letter and Author response

Decision letter https://doi.org/10.7554/eLife.77146.sa1
Author response https://doi.org/10.7554/eLife.77146.sa2

---

# Additional files

## Supplementary files

- Transparent reporting form
- Reporting standard 1. STROBE Checklist.

## Data availability

All de-identified contrast imaging data and variables, involving the documentation of the processing procedure are available: Basedau, Hauke et al. (2022), Migraine monoclonal antibodies against CGRP change brain activity depending on ligand or receptor target: a functional magnetic resonance imaging study, Dryad, Dataset, https://doi.org/10.5061/dryad.w3r2280t2.

The following dataset was generated:

| Author(s) | Year | Dataset title | Dataset URL | Database and Identifier |
|-----------|------|---------------|-------------|-------------------------|
| Basedau H, Sturm L, Mehnert J, Peng K, Schellong M, May A | 2022 | Migraine monoclonal antibodies against CGRP change brain activity depending on ligand or receptor target: a functional magnetic resonance imaging study | https://doi.org/10.5061/dryad.w3r2280t2 | Dryad Digital Repository, 10.5061/dryad.w3r2280t2 |

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
