## [Editor Report]

Antibodies to CGRP or to the CGRP receptor are now approved for the management of migraine. Interestingly, although antibodies are presumed not to penetrate the blood-brain barrier, consistent with findings previously shown for antibodies directed against the receptor, the authors now report that galcanezumab, which targets the CGRP peptide, decreases hypothalamic activation, more so in responders. They suggest that these CNS actions may underlie part of the efficacy of these drugs in the management of migraine.

---

## [Decision Letter]

**Decision letter after peer review:**

Thank you for submitting your article "Migraine monoclonal antibodies against CGRP change brain activity depending on ligand or receptor target: a functional magnetic resonance imaging study" for consideration by *eLife*. Your article has been reviewed by 3 peer reviewers, and the evaluation has been overseen by a Reviewing Editor and Jeannie Chin as the Senior Editor. The following individual involved in the review of your submission has agreed to reveal their identity: Andrew Charles (Reviewer #3).

1) The Reviewers have made several suggestions that should be addressed in revising your manuscript. Also it would be very helpful to elaborate on the interesting distinction between the effect of galcanezumab and erenumab.

*Reviewer #1 (Recommendations for the authors):*

The paper reports a well conducted, carefully analyzed data set with important implications for understanding migraine therapeutics. My comments are modest and advisory.

A. The Introduction states CGRP mABs are impermeable with regard to the blood-brain barrier. An article is cited that, consistent with what is known, shows their permeability is limited not zero (1). The authors could consider recasting the sentence in line with the citation.

B. Not all responders to galcanezumab will do so in the first two weeks (2); would this have blunted the responder/non-responder comparison at the two week point? Could it be argued that a longer study time might be considered going forward?

C. The authors cite six papers for the Method; perhaps one or two would be sufficient?

D. Given the highly personalized nature of pain, it could be argued that VAS scales by virtue of the underlying biological variability between patients, are poor candidates for being treated as more than within subject ordinal measures? It could be argued that the Wilcoxon test is preferable as has been mentioned by the authors.

E. Given response rates to galcanezumab vary by frequency of migraine days, as evidenced by headline responses in episodic (3, 4) and chronic migraine (5), did migraine day frequency effect outcomes? Tables one might usefully offer the frequency data for both the episodic and chronic migraine groups.

F. Regarding concomitant preventives, it is a practical limitation of studies that stopping preventives can be problematic. If the patients taking concomitant preventives with established efficacy in migraine are excluded, and not all those listed un characteristics would fulfill that criterion, does this alter the outcome?

G. It would be generally accepted that a 30% reduction in migraine days in episodic migraine is a modest effect; if a 50% reduction had been used as the binary response outcome for all participants, would the result be different? The authors very sensibly point out the severity change argument in the limitations section. It could be argued that applying the 50% rule might sharpen the differences.

References

1. Johnson KW, Morin SM, Wroblewski VJ, Johnson MP. Peripheral and central nervous system distribution of the CGRP neutralizing antibody [(125)I] galcanezumab in male rats. Cephalalgia. 2019;39:1241-8.

2. Goadsby PJ, Dodick DW, Martinez JM, Ferguson MB, Oakes TM, Zhang Q, et al., Onset of Efficacy and Duration of Response of Galcanezumab for the Prevention of Episodic Migraine: A Post-Hoc Analysis. Journal of Neurology, Neurosurgery and Psychiatry. 2019;90:939-44.

3. Skljarevski V, Matharu M, Millen BA, Ossipov MH, Kim BK, Yang JY. Efficacy and safety of galcanezumab for the prevention of episodic migraine: Results of the EVOLVE-2 Phase 3 randomized controlled clinical trial. Cephalalgia. 2018;38:1442-54.

4. Stauffer VL, Dodick DW, Zhang Q, Carter JN, Ailani J, Conley RR. Evaluation of Galcanezumab for the Prevention of Episodic Migraine: The EVOLVE-1 Randomized Clinical Trial. JAMA Neurol. 2018;75:1080-8.

5. Detke HC, Goadsby PJ, Wang S, Friedman DI, Selzler K, Aurora SK. Galcanezumab in chronic migraine. The randomized, double-blind, placebo-controlled REGAIN study. Neurology (Minneap). 2018;91:e2211-e21.

*Reviewer #2 (Recommendations for the authors):*

In this study, Basedau et al., seek to further understand the mechanism of action of CGRP monoclonal antibodies for migraine. These drugs likely do not gain substantial access to the CNS given their size, yet they show efficacy for a neurological disorder that has numerous aspects mediated by the CNS. Thus, the exact mechanism by which they treat migraine is not clear. This group performed a similar study recently using a distinct CGRP antibody that binds to the receptor (erenumab) and the current study extends this work using a CGRP antibody that binds to the peptide (galcanezumab). Using fMRI in 26 migraine patients, the authors performed scans to assess brain activity following nasal administration of ammonia, rose odor, or visual checkerboard stimulation both prior to and three weeks following galcanezumab treatment. Unpleasantness ratings were collected and headache diaries were compiled for each patient, including efficacy of the drug for migraine attacks during the study. Approximately 70% of the patients reported efficacy of galcanezumab for migraine attacks and were defined as responders to the drug. Despite efficacy for headache in the majority of subjects, there were no differences in intensity or unpleasantness ratings for any of the stimuli following drug treatment compared to prior. In contrast, galcanezumab treatment decreased trigeminal stimuli-mediated activation of a number of CNS regions, including the hypothalamus, in all subjects. The regions where stimulated activation were reduced by galcanezumab showed distinct patterns when broken down by responder vs non-responder groups with respect to drug efficacy for migraine attacks. Additionally, when the new data with galcanezumab were compared with prior data generated using erenumab, distinct drug-induced changes in activation patterns were observed. Finally, there was a highly co-variant relationship between the effect of galcanezumab on trigeminal stimulation mediated activation patterns and headache efficacy for responses within the spinal trigeminal nucleus (STN) and functional connectivity of the STN. This latter finding was the basis of a conclusion by the authors that response patterns in the STN could be predictive and/or required for galcanezumab efficacy for migraine. In general, the studies add to the understanding of how CGRP monoclonal antibodies influence CNS responses to painful stimuli.

The manuscript is well written and the conclusions are supported by the data. There were some questions and comments identified related to the study.

1. The authors suggest in the discussion that galcanezumab binding to peripheral CGRP could subsequently influence the circulating levels of CGRP and as a result, modulate its ability to cross the blood brain barrier and modulate hypothalamic function. While this is true for galcanezumab, this is unlikely to be true for erenumab since it binds the receptor. Is it possible that these types of differences are responsible for the distinct patterns of effects observed when comparing the two antibodies? The authors discuss other possible explanations in the paper (starting at line 344) but this.

2. Given that effects of gancanezumab were observed in the hypothalamus, and the hypothalamus has areas not protected by the blood brain barrier, could the observed effects on this region be due to direct access of drug to this CNS region? It might be helpful to discuss the specific regions of the hypothalamus not protected by the BBB and whether these could contribute to the observed effects. This was included in the prior erenumab paper but would be helpful to discuss here.

3. The prior erenumab study included 81% females while the current study is 96% female. While not a major difference, do the authors expect this has any influence on the findings of the two studies?

4. It might be helpful to include more detail on the information collected in the headache calendars. This may help readers assess the level of attention subjects paid to the qualities of their migraines across the 3 months. Since this study uses drug efficacy as a method to segregate patients responses to trigeminal stimulation, and efficacy was entirely subjective on the part of the subjects, it seems helpful to have more detail on what was asked of the patients between study visits.

The current work is quite similar to the prior study by these same authors using erenumab. It would help increase the impact for *eLife* if the authors could provide more information (or data) that clarifies what the major advance is in this work over the prior work.

*Reviewer #3 (Recommendations for the authors):*

It is now well established that CGRP plays a fundamental role in migraine, and therapies targeting CGRP can be highly effective in reducing frequency and severity of migraine. However, multiple aspects of the specific mechanisms of CGRP in migraine remain poorly understood. Ongoing questions include: What triggers the release of CGRP that is involved in migraine? Where does migraine-related CGRP release occur, and where does CGRP binding mediate migraine-related effects? Are the therapeutic effects of CGRP-targeted treatments mediated primarily by the peripheral or central nervous system, or both?

The current study aims to shed light on the potential therapeutic mechanisms of CGRP-targeted treatments by using fMRI to examine changes in brain activity and connectivity before and after treatment with galcanezumab. These results are compared with a previous similar study of erenumab. The results clearly show central effects of the antibodies, notably in the hypothalamus, and a comparison of the two treatments suggests that there are distinctive patterns of reduction in brain activity resulting from each of the two therapies. The results also show that activation of the spinal trigeminal nucleus by noxious stimulation is correlated with the response to therapy. The results are an important description of the brain effects CGRP-targeted antibodies. Although it is widely presumed that because of their size these antibodies must be working peripherally, the studies do not rule out the possibility that they could be acting in brain regions (like the hypothalamus) that are outside of the blood brain barrier. Although the present study does not definitively answer the question regarding the site(s) of action of the CGRP-targeted therapies, it provides key information about central effects that will enable future studies to address this question. In addition, it identifies patterns of brain activity that could at some point become "biomarkers" of treatment response.

The imaging studies are of high quality, and the experimental design importantly enables within-subject comparison before and after treatment.

There are, however, some sources of variability that require consideration in the interpretation of the results, and some interesting findings that warrant further discussion.

1. The study includes both episodic and chronic migraine patients who are on a variety of different medications. In order to reduce potential migraine phase-related variability, a subgroup of subjects who had either headache or non headache on scan days both pre and post treatment. As it turned out , 14/15 of the phase-consistent patients were scanned on a headache day pre- and post-treatment with galcanezumab, while only 1 was scanned on non-headache days. The fact that nearly all patients were scanned on headache days vs. non-headache days could have important effects on the results. It is also unclear how many of the phase-consistent patients had chronic migraine vs. episodic migraine, which could be another important contributing variable.

2. The responder rates for both the current galcanezumab study and the previously reported erenumab study that is used for comparison are significantly less than those reported in the clinical trials of these therapies. This suggests that the subjects examined in these studies may be different from the general clinical trial population.

3. The authors report that there was no difference in headache severity on the day of the scan pre- and post- treatment, despite reductions in brain activation in different regions, particularly in those who responded to treatment with reduced attack severity. This suggests that the reduction in brain activation is not correlated with attack severity, in contrast to the correlation with attack frequency.

4. The unpleasantness perception in response to trigeminal nociceptive stimulation apparently did not change after treatment despite reductions in the brain responses seen with fMRI. Here again there is a dissociation between the acute clinical response and the changes in brain activity.

5. Regarding the comparison of the erenumab data vs. the galcanezumab data, there is some concern regarding the comparison of these two groups. Although the demographics of the groups are generally similar, it is not clear how many of the phase-consistent scans in the erenumab study were performed on headache days vs. non-headache days, and how many were in chronic vs. episodic migraine patients. These details would be important to know in considering whether the comparison between the two groups is feasible.

6. Since the subjects were primarily women, another potential source of variability is the menstrual cycle. Is there any indication that this could be an issue regarding the interpretation of the data?

7. Regarding the conclusions, the authors should acknowledge the possibility that the antibodies could be acting directly in the hypothalamus, parts of which may be outside of the blood brain barrier. Some brain areas including the hypothalamus have been shown to express CGRP receptors. (Eftekhari S, Gaspar RC, Roberts R, et al., Localization of CGRP receptor components and receptor binding sites in rhesus monkey brainstem: A detailed study using in situ hybridization, immunofluorescence, and autoradiography. The Journal of comparative neurology. 2016;524:90-118).

8. Please indicate the number of subjects compared in the phase-consistent studies of galcanezumab and erenumab who had episodic vs. chronic migraine.

9. For the erenumab study that is compared, please indicate the number of scans in the phase-consistent studies that were performed on headache dys vs. non-headache days.

10. Please comment on the lack of correlation between the clinical measures of headache severity or unpleasantness of the noxious stimulation and the imaging findings on MRI in terms of the reduction in brain activity in response to galcanezumab.

11. A brief discussion of the possibility that galcanezumab (and erenumab) could be exerting effects via brain regions like the hypothalamus that may be outside the blood brain barrier is warranted.

12. p. 17, line 339 -- I think the authors mean "express" instead of "expose".

13. It would be helpful to have a greater description in the figure legends for the results tables or elsewhere regarding what each of the values mean and how they were calculated. For example, I am unclear as to how the T values were calculated.

---

## [Author Response]

Reviewer #1 (Recommendations for the authors):The paper reports a well conducted, carefully analyzed data set with important implications for understanding migraine therapeutics. My comments are modest and advisory.A. The Introduction states CGRP mABs are impermeable with regard to the blood-brain barrier. An article is cited that, consistent with what is known, shows their permeability is limited not zero (1). The authors could consider recasting the sentence in line with the citation.

Thank you for the suggestion; we have changed the statement accordingly (p. 4 line 5f.).

B. Not all responders to galcanezumab will do so in the first two weeks (2); would this have blunted the responder/non-responder comparison at the two week point? Could it be argued that a longer study time might be considered going forward?

Good point. For this reason we took the subjective assessment of the participants after 3 weeks as well as the headache calendar after a total of 3 months of therapy into account. This interval is also specified in the German guideline on CGRP therapy for migraine (p. 6 line 16f. and p. 10 line 7). Interestingly, the outcome of the 2 time points was highly correlated.

C. The authors cite six papers for the Method; perhaps one or two would be sufficient?

We have reduced the number of citations in the methods section

D. Given the highly personalized nature of pain, it could be argued that VAS scales by virtue of the underlying biological variability between patients, are poor candidates for being treated as more than within subject ordinal measures? It could be argued that the Wilcoxon test is preferable as has been mentioned by the authors.

We totally agree. We changed the wording accordingly. Indeed we used a NRS (anchored at 0 – 100). NRS should be treated as a continuous variable and we therefore used paired t-tests.

E. Given response rates to galcanezumab vary by frequency of migraine days, as evidenced by headline responses in episodic (3, 4) and chronic migraine (5), did migraine day frequency effect outcomes? Tables one might usefully offer the frequency data for both the episodic and chronic migraine groups.

We conducted sub-analyses of the functional imaging data as well as the analyses mentioned in the manuscript with the diagnosis (episodic vs. chronic) as a covariate. No significant influence by migraine diagnosis could be found. The frequencies for episodic vs. chronic are now given in the manuscript (p.11 line 11 ) as well as in table 1.

F. Regarding concomitant preventives, it is a practical limitation of studies that stopping preventives can be problematic. If the patients taking concomitant preventives with established efficacy in migraine are excluded, and not all those listed un characteristics would fulfill that criterion, does this alter the outcome?

Concomitant preventives may well influence the results, especially when there is any change (including dosage) between T0 and T1. For this reason, concomitant preventives are only allowed where the kind and the dosage of any medication have remained unchanged starting 3 months before and throughout the entire study period.

G. It would be generally accepted that a 30% reduction in migraine days in episodic migraine is a modest effect; if a 50% reduction had been used as the binary response outcome for all participants, would the result be different? The authors very sensibly point out the severity change argument in the limitations section. It could be argued that applying the 50% rule might sharpen the differences.

We absolutely agree with the reviewer on this point. We therefore also carried out this analysis with 50% responders although not many patients qualified, and found the same result at a less stringent (explorative) significance level (p. 14 line 15f.) because of lower patient numbers.

References1. Johnson KW, Morin SM, Wroblewski VJ, Johnson MP. Peripheral and central nervous system distribution of the CGRP neutralizing antibody [(125)I] galcanezumab in male rats. Cephalalgia. 2019;39:1241-8.2. Goadsby PJ, Dodick DW, Martinez JM, Ferguson MB, Oakes TM, Zhang Q, et al., Onset of Efficacy and Duration of Response of Galcanezumab for the Prevention of Episodic Migraine: A Post-Hoc Analysis. Journal of Neurology, Neurosurgery and Psychiatry. 2019;90:939-44.3. Skljarevski V, Matharu M, Millen BA, Ossipov MH, Kim BK, Yang JY. Efficacy and safety of galcanezumab for the prevention of episodic migraine: Results of the EVOLVE-2 Phase 3 randomized controlled clinical trial. Cephalalgia. 2018;38:1442-54.4. Stauffer VL, Dodick DW, Zhang Q, Carter JN, Ailani J, Conley RR. Evaluation of Galcanezumab for the Prevention of Episodic Migraine: The EVOLVE-1 Randomized Clinical Trial. JAMA Neurol. 2018;75:1080-8.5. Detke HC, Goadsby PJ, Wang S, Friedman DI, Selzler K, Aurora SK. Galcanezumab in chronic migraine. The randomized, double-blind, placebo-controlled REGAIN study. Neurology (Minneap). 2018;91:e2211-e21.

Thank you, all references cited, where appropriate

Reviewer #2 (Recommendations for the authors):The manuscript is well written and the conclusions are supported by the data. There were some questions and comments identified related to the study.1. The authors suggest in the discussion that galcanezumab binding to peripheral CGRP could subsequently influence the circulating levels of CGRP and as a result, modulate its ability to cross the blood brain barrier and modulate hypothalamic function. While this is true for galcanezumab, this is unlikely to be true for erenumab since it binds the receptor. Is it possible that these types of differences are responsible for the distinct patterns of effects observed when comparing the two antibodies? The authors discuss other possible explanations in the paper (starting at line 344) but this.

The reviewer is exactly right. We have discussed this issue. Relevant discussion is highlighted in in the text (see page 17).

2. Given that effects of gancanezumab were observed in the hypothalamus, and the hypothalamus has areas not protected by the blood brain barrier, could the observed effects on this region be due to direct access of drug to this CNS region? It might be helpful to discuss the specific regions of the hypothalamus not protected by the BBB and whether these could contribute to the observed effects. This was included in the prior erenumab paper but would be helpful to discuss here.

Thank you for this comment. We have now included this line of thought into the discussion (page 17)

3. The prior erenumab study included 81% females while the current study is 96% female. While not a major difference, do the authors expect this has any influence on the findings of the two studies?

Thank you for this comment. We agree that sex hormones play a great role. The difference between the 2 studies (3 males) is however too small to explain the difference in imaging results. Nevertheless, following your suggestion we also carried out a sub-analysis in this regard with only the female study participants and found the same result.

4. It might be helpful to include more detail on the information collected in the headache calendars. This may help readers assess the level of attention subjects paid to the qualities of their migraines across the 3 months. Since this study uses drug efficacy as a method to segregate patients responses to trigeminal stimulation, and efficacy was entirely subjective on the part of the subjects, it seems helpful to have more detail on what was asked of the patients between study visits.

The classification of the responder group and non-responder group was made after an interval of 3 months with the help of a standardised headache calendar (p. 6 line 16- https://www.dmkg.de/files/Kopfschmerzkalender_PDF/Kopfschmerzkalender_ENGLISCH_18.3.2021.pdf). We took the subjective assessment of the participants after 3 weeks as well as the headache calendar after a total of 3 months of therapy into account. This interval is also specified in the German guideline on CGRP therapy for migraine (p. 6 line 17 and p. 10 line 7). Interestingly, the outcome of the 2 time points were highly correlated. (p. 10 line 7).

The current work is quite similar to the prior study by these same authors using erenumab. It would help increase the impact for eLife if the authors could provide more information (or data) that clarifies what the major advance is in this work over the prior work.

The main advance of the current work compared to our previous study using erenumab lies in the mechanism of how the medication works. Different from erenumab, which targets the CGRP receptor, galcanezumab directly targets the peptide itself. For this reason, we additionally analysed the raw data from both studies (i.e. did not just take analysed data from a historical patient group), and such direct comparisons are therefore in a strong position to point out differences and similarities between the ligand and the receptor antibody on central pain modulation in migraine patients. We additionally and for the first time show that the STN, which plays a central role in the ascending trigemino-nociceptive pathway, co-varies in its activity already at visit 1 with the response, i.e. the efficacy to galcanezumab. These data are new and demonstrate, using functional imaging, a predictor for clinical efficacy of galcanezumab.

Reviewer #3 (Recommendations for the authors):The imaging studies are of high quality, and the experimental design importantly enables within-subject comparison before and after treatment.There are, however, some sources of variability that require consideration in the interpretation of the results, and some interesting findings that warrant further discussion.1. The study includes both episodic and chronic migraine patients who are on a variety of different medications. In order to reduce potential migraine phase-related variability, a subgroup of subjects who had either headache or non headache on scan days both pre and post treatment. As it turned out , 14/15 of the phase-consistent patients were scanned on a headache day pre- and post-treatment with galcanezumab, while only 1 was scanned on non-headache days. The fact that nearly all patients were scanned on headache days vs. non-headache days could have important effects on the results. It is also unclear how many of the phase-consistent patients had chronic migraine vs. episodic migraine, which could be another important contributing variable.

We apologise for the misleading wording in the manuscript. The number of patients with headache on day 1 and day 2 referred to the total cohort. Overall, the ratio of no headache on both days (n=8) and headache on both days (n=7) is balanced, especially as we have evaluated the imaging data intraindividually and longitudinally. We have clarified this in the manuscript (see p. 12 line 4f)

2. The responder rates for both the current galcanezumab study and the previously reported erenumab study that is used for comparison are significantly less than those reported in the clinical trials of these therapies. This suggests that the subjects examined in these studies may be different from the general clinical trial population.

True. The participants fulfilled the official criteria for CGRP therapy according to the German guideline. This guideline requires that all oral prophylactics have already been tried unsuccessfully, and, in case of chronic migraine patients, have also failed botulinum toxin treatment. Thus, the patients included here formally correspond to a patient group that is difficult to treat (p. 13 line 4). We would argue that the principles of mechanism how the respective antibodies exert their effect (in this case centrally) are however the same in treatment naïve and non-responding patients. We agree that this needs to be investigated in future studies to give a valid answer.

3. The authors report that there was no difference in headache severity on the day of the scan pre- and post- treatment, despite reductions in brain activation in different regions, particularly in those who responded to treatment with reduced attack severity. This suggests that the reduction in brain activation is not correlated with attack severity, in contrast to the correlation with attack frequency.

That is an interesting thought. We used the (quantifiable) reduction in the number of monthly headache days to evaluate the efficacy of migraine prophylaxis. The subgroup analysis (same headache state between T0 and T1) that this reviewer mentions was indeed controlled for headache severity, and no difference between headache severity on that special day was found. This allowed making the assumption, that our results are frequency dependant.

We agree that over the whole time span of 3 months the mean severity (and other subjective measures) may well be reduced nevertheless. We cannot more than agree that our findings therefore may be restricted to patients who indeed are frequency responders. Since in our experience many patients report that indeed the severity is reduced more than frequency, one could argue that only in frequency responders the hypothalamus is effectively targeted using mABs and not in severity responders. Unfortunately, we had not enough patients who kept the frequency and only reported a severity decrease, what would have been necessary to directly answer that question. (we have added this in the discussion please see page 19)

4. The unpleasantness perception in response to trigeminal nociceptive stimulation apparently did not change after treatment despite reductions in the brain responses seen with fMRI. Here again there is a dissociation between the acute clinical response and the changes in brain activity.

Our remark regarding unpleasantness perception is explicitly for the response to (experimental) trigeminal nociceptive stimulation. Since CGRP monoclonal antibody are not pain killers, we would not expect a difference in perceiving this experimental stimulus and consequently controlled for this.

5. Regarding the comparison of the erenumab data vs. the galcanezumab data, there is some concern regarding the comparison of these two groups. Although the demographics of the groups are generally similar, it is not clear how many of the phase-consistent scans in the erenumab study were performed on headache days vs. non-headache days, and how many were in chronic vs. episodic migraine patients. These details would be important to know in considering whether the comparison between the two groups is feasible.

We apologise for the misleading wording in the manuscript. The number of patients with headache on day 1 and day 2 referred to the total cohort. Overall, the ratio of no headache on both days (n=8) and headache on both days (n=7) is balanced, especially as we have evaluated the imaging data intraindividually and longitudinally. We have clarified this in the manuscript (see p. 12 line 4 and Table 1)

6. Since the subjects were primarily women, another potential source of variability is the menstrual cycle. Is there any indication that this could be an issue regarding the interpretation of the data?

We agree that the menstrual cycle could have an impact in general. However, In our previous study, we scanned seven patients daily over 30 days using the same experimental set-up and did not find changes in imaging results in regards to the menstrual cycle. Therefore, we did not control for the menstrual cycle in the current study[1].

1. Schulte, L. H., Mehnert, J. and May, A. Longitudinal Neuroimaging over 30 Days: Temporal Characteristics of Migraine. Annals of Neurology 87, 646–651 (2020).

7. Regarding the conclusions, the authors should acknowledge the possibility that the antibodies could be acting directly in the hypothalamus, parts of which may be outside of the blood brain barrier. Some brain areas including the hypothalamus have been shown to express CGRP receptors. (Eftekhari S, Gaspar RC, Roberts R, et al., Localization of CGRP receptor components and receptor binding sites in rhesus monkey brainstem: A detailed study using in situ hybridization, immunofluorescence, and autoradiography. The Journal of comparative neurology. 2016;524:90-118).

Thank you for this remark, we have now discussed this fact and cited the reference (page 17 line 19ff)

8. Please indicate the number of subjects compared in the phase-consistent studies of galcanezumab and erenumab who had episodic vs. chronic migraine.

Done (see table 1)

9. For the erenumab study that is compared, please indicate the number of scans in the phase-consistent studies that were performed on headache dys vs. non-headache days.

Done (see p. 12 line 5ff)

10. Please comment on the lack of correlation between the clinical measures of headache severity or unpleasantness of the noxious stimulation and the imaging findings on MRI in terms of the reduction in brain activity in response to galcanezumab.

That is an interesting thought. We used the (quantifiable) reduction in the number of monthly headache days to evaluate the efficacy of migraine prophylaxis. The subgroup analysis (same headache state between T0 and T1) that this reviewer mentions was indeed controlled for headache severity, and no difference between headache severity on that special day was found. This allowed making the assumption, that our results are frequency dependant.

We agree that over the whole time span of 3 months the mean severity (and other subjective measures) may well be reduced nevertheless. We cannot more than agree that our findings therefore may be restricted to patients who indeed are frequency responders. Since in our experience many patients report that indeed the severity is reduced more than frequency, one could argue that only in frequency responders the hypothalamus is effectively targeted using mABs and not in severity responders. Unfortunately, we had not enough patients who kept the frequency and only reported a severity decrease, what would have been necessary to directly answer that question. (we have added this in the discussion please see page 19)

11. A brief discussion of the possibility that galcanezumab (and erenumab) could be exerting effects via brain regions like the hypothalamus that may be outside the blood brain barrier is warranted.

Thank you for this remark, we have now discussed this fact and cited the reference (page 17 line 19ff)

12. p. 17, line 339 -- I think the authors mean "express" instead of "expose".

We corrected this.

13. It would be helpful to have a greater description in the figure legends for the results tables or elsewhere regarding what each of the values mean and how they were calculated. For example, I am unclear as to how the T values were calculated.

We explicitly state an extended description of the individual statistical results by exemplifying the exact comparison, the number of participants (n), the degrees of freedom, the p-values and t-values for all t-tests. We calculated T-test with the standard t-test family implemented in the SPM toolbox (Wellcome Trust Center for Neuroimaging, London, UK) SPM12 – Statistical Parametric Mapping. https://www.fil.ion.ucl.ac.uk/spm/software/spm12/ (accessed 4 Jan 2019).

We now also include a short description of the used test in the main text table, as well as in the Table land Figure legends.